# The Expression Characteristics and Function of the RECQ Family in Pan-Cancer

**DOI:** 10.3390/biomedicines11082318

**Published:** 2023-08-21

**Authors:** Yuanyuan Zhou, Xucheng Huang, Liya Wang, Yujia Luo

**Affiliations:** 1Department of Reproductive Endocrinology, Women’s Hospital, School of Medicine, Zhejiang University, Hangzhou 310003, China; zhouyuanyuan0624@zju.edu.cn (Y.Z.); wangliya@zju.edu.cn (L.W.); 2Department of Clinical Laboratory, Sir Run Run Shaw Hospital, School of Medicine, Zhejiang University, Hangzhou 310016, China; tracyhuang@zju.edu.cn; 3Department of NICU, Sir Run Run Shaw Hospital, School of Medicine, Zhejiang University, Hangzhou 310016, China

**Keywords:** RECQ family, tumor immunity, pan-cancer, DNA repair, multi-omics bioinformatics

## Abstract

Background: The genes of the RECQ DNA helicase family play a part in preserving the stability of the genome and controlling different disease mechanisms. However, the expression features of RECQs in relation to pan-cancer, their correlation with the immune microenvironment of tumors, and the landscape of prognostic power are still undisclosed. Methods: Various sequence and clinical data extracted from 33 cancers were utilized to generate a comprehensive overview of RECQs in the landscape. Afterward, we discovered variations in gene expression, potential enrichment of functions, genetic alterations, and analysis related to the immune response in tumors. Additionally, we explored the clinical characteristics and diagnostic significance of RECQs. And the important association of RECQL4 with liver hepatocellular carcinoma (LIHC) was investigated. Results: RECQs exhibited extensive mutations in different types of cancers. The expression of RECQ may be influenced by an oncogenic mutation in certain types of cancer, resulting in the observed genomic and epigenetic changes in diverse tumor formations. Furthermore, RECQs originating from tumors exhibited a significant association with the immune microenvironment of the tumor, indicating their potential as promising targets for therapy. Patient prognosis was significantly associated with the majority of genes in the RECQ family. In LIHC, RECQL4 eventually emerged as a separate prognostic determinant. Conclusions: To summarize, RECQs are essential for the regulation of the immune system in tumors, and RECQL4 serves as a prognostic indicator in LIHC. The results of our study offer fresh perspectives on RECQs from a bioinformatics perspective and emphasize the importance of RECQs in the diagnosis and treatment of cancer.

## 1. Introduction

The detrimental effects of exposure to toxic chemicals, oxidation, free radicals, ultraviolet light, and ionizing radiation on human DNA, specifically in the form of double-stranded breaks, pose significant challenges in terms of repair [1,2]. The genome may become destabilized, potentially resulting in cancer, due to faulty signals for damage response and DNA that remains unrepaired [3]. In addition to radiotherapy and chemotherapy, therapies that enhance the ability body’s immunity to fight against tumors have become the accepted practice for individuals with cancer in recent times [4,5,6]. Despite the notable efficacy demonstrated by targeted immunotherapies such as CAR-T (Chimeric antigen receptor T) cell, anti-cytotoxic T lymphocyte-associated antigen-4 (CTLA4), and anti-Programmed death-1 (PD1)/anti-PDL1 therapy, it is important to note that immunotherapy is effective only for a specific group of patients. To effectively tackle both primary and acquired resistance, it is crucial to employ a variety of therapeutic strategies [7,8,9,10]. Hence, appropriate predictive indicators and treatment objectives should be discovered urgently.

The DNA damage response, a collection of mechanisms, has been developed by eukaryotic cells to identify, transmit signals, and mend DNA damage [3]. The RECQ group of DNA helicases is among the most extensively preserved helicase families during the process of evolution and plays a crucial role in maintaining the overall integrity of the genome [11]. Five RECQ helicase proteins, namely RECQL, BLM, WRN, RECQL4, and RECQL5, have been identified in the human species. RECQ helicases attach to and move along a single DNA strand in a 3′→5′ orientation, and single-stranded DNA enhances their ability to unwind double-stranded DNA. RECQ helicases play a crucial role in all DNA processes, such as recombination, replication, and DNA repair, by separating double-stranded nucleic acids [12]. Various human diseases, such as genetic disorders, diseases caused by the environment, aging, and tumors, are linked to faulty mechanisms for repairing damaged DNA in the organism.

Understanding the impact of RECQs on the development and progression of cancer is crucial as chromosomal instability plays a significant role in the onset of cancer. Researchers have shown significant interest in the involvement of RECQs in cancer over the years. Abnormal expression of RECQs has been related to the progression of different cancers and is correlated with unfavorable prognosis [13,14]. Recent studies indicate that RECQs may serve as potential genes responsible for breast cancer susceptibility, with missense mutations in these genes playing a role in the development of familial breast cancer [15,16,17]. In addition, RECQs are linked to thyroid carcinoma, prostate cancer, and ovarian cancer [18,19]. RECQs play a wide-ranging and varied role in the regulation of cancer and are closely associated with renowned molecules or pathways that control cancer, thereby having significant implications for cancer treatment [20]. This implies that RECQs present a compelling opportunity for the diagnosis and treatment of tumors. However, the immune function, the level of expression, and the predictive significance of RECQs in pan-cancer have not been thoroughly investigated.

In this study, we extensively investigated the genomic, transcriptomic, immune microenvironment, prognostic value, and therapeutic implications for the RECQ family across various types of cancer. In liver hepatocellular carcinoma (LIHC), we found RECQL4 as an important determinant of prognosis. The results of our study will offer supplementary insights into the importance of RECQs, leading to new opportunities for enhancing immunotherapy and precisely forecasting prognosis.

## 2. Methods

### 2.1. Source of Data and Its Processing

The Cancer Genome Atlas Program (TCGA), which can be accessed at https://portal.gdc.cancer.gov/ (accessed on 1 March 2023), is an open database containing Next Generation Sequencing data from more than 11,000 tumor samples across 33 different types of cancers. These samples provided gene expression, copy number alterations, methylation, and clinical data. Methylation data were analyzed using the Illumina human methylation 450 platform. The Genotype-Tissue Expression (GTEx) database, which provides publicly available expression data from approximately 1000 individuals with 53 normal sample sites, was utilized for this investigation. UCSC Xena (https://xenabrowser.net/datapages/ (accessed on 1 March 2023)) provided the GTEx and TCGA RNAseq data, which were utilized for the analysis of unpaired samples in the TCGA-GTEx combined study [21]. R limma package was applied to integrate and process TCGA and GTEx data [22]. In addition, data from the Gene Expression Omnibus (GEO) dataset GSE14520 was analyzed to verify the gene expression of paired samples in cancer patients.

### 2.2. Analysis of RECQs for Mutations and CNVs

The cBioPortal database (http://www.cbioportal.org (accessed on 15 March 2023)), an accessible resource tool for analyzing and visualizing various genomic data related to cancer, was utilized to assess the mutation rates of RECQs in 33 different tumor types [23,24]. The TCGA Pan-Cancer Atlas encompasses a total of 10,519 samples in this comprehensive study. The Gene Set Cancer Analysis is a database and it collects data on cancer genomes and gene sets related to immunogenomics [25]. The RECQ family’s copy number variations (CNV) in various cancers from the TCGA were examined during our investigation.

### 2.3. Evaluation of the Level of mRNA and Protein Expression

Cancer Cell Line Encyclopedia (CCLE) database was performed to obtain mRNA expression data for 10 LIHC cell lines (https://sites.broadinstitute.org/ccle/ (accessed on 30 March 2023)). The Human Protein Atlas (HPA) database is a freely available and transparent platform offering researchers valuable insights into the expression and localization of proteins in various human tissues, cells, and organs [26,27]. The HPA database was utilized to examine the protein expression level of the RECQ family in different types of cancers through immunohistochemistry analysis. Antibodies specific to the target were used to obtain immunocytochemistry images for the detection and visualization of RECQL4 protein in the human U251MG glioblastoma cell line and A431 epidermoid carcinoma cell line.

### 2.4. Building Networks of Protein Interactions

The String database compiles an extensive volume of publicly accessible data pertaining to protein–protein interactions (PPI) [28]. The RECQs PPI network was built using the String database. The ggraph package in R was applied to analyze and observe the interactive network [29].

### 2.5. Perform Enrichment Analysis Using GO and KEGG

Proteins that interacted with RECQs were chosen from the String database and examined for enrichment in Gene Ontology (GO) and Kyoto Encyclopedia of Genes and Genomes (KEGG). Additionally, the clusterProfiler software was employed to investigate biological processes (BP), cellular component (CC), molecular function (MF), and KEGG pathway items that showed significant enrichment, with all displayed items having a *p*-value < 0.05 [30,31].

### 2.6. Analysis Related to the Immune System

We acquired data on pan-cancer immune subtypes from TCGA and utilized the Kruskal test to assess the disparities. The tumor-immune system interactions database was utilized to assess the immune subtype in every type of cancer, encompassing C1 (wound healing), C2 (IFN-gamma dominant), C3 (inflammatory), C4 (lymphocyte depleted), C5 (immunological quiet), and C6 (TGF-beta dominant) [32]. The ImmuneScore, StromalScore, ESTIMATEScore, RNAss, DNAss, tumor mutational burden (TMB), and microsatellite instability (MSI) were computed using the TCGA tumor samples. A Spearman correlation analysis was conducted by combining these data with RECQ family gene expression data. The Tumor Immune Estimation Resource (TIMER) algorithm was employed to investigate the correlation between tumor-infiltrating immune cells (TICs) and RECQs [33].

### 2.7. Survival Analysis

Patient clinical data were acquired from the TCGA repository. Afterward, the patients were categorized into groups with high and low levels of gene expression, using the median as a reference point. This categorization was completed in order to examine the overall survival (OS), disease-specific survival (DSS), disease-free interval (DFI), and progression-free interval (PFI) of the patients. Moreover, the prognostic significance of RECQs mRNA expression in various cancers was investigated using the Kaplan–Meier plotter database [34]. In order to determine the statistical significance, a hazard ratio (HR) with a ninety-five percent confidence interval (CI) and a log-rank *p*-value were calculated, considering *p* < 0.05. Statistical analysis was conducted using the survival package. We utilized the survminer package to generate Kaplan–Meier plots and for data visualization. Using the rms package, a nomogram was utilized to generate predictions for mortality rates and create a calibration plot to compare actual and model probabilities across various scenarios.

### 2.8. Statistical Analysis

For comparisons between groups, the Wilcoxon rank sum test was performed, specifically for comparing two paired samples or ordered classification variables. Gene expression was correlated using Spearman analysis. The chi-square test or Fisher’s exact test was used to compare the clinical feature differences between groups. The diagnostic significance of RECQs in LIHC was indicated by the area beneath the receiver operating characteristic (ROC) curve. Survival analyses were performed using multivariate and univariate Cox regression models, and multivariate analysis included variables with a *p* value < 0.05 in univariate analysis. All statistical analyses were finished using R. Image visualization was performed using the ggplot2 library. A *p*-value < 0.05 was used to define a statistically significant distinction.

## 3. Results

### 3.1. Analysis of RECQs’ Expression, Interaction, and Functional Enrichment

Initially, the co-expression of RECQs on the genome level was investigated in pan-cancer, revealing a robust association between BLM and RECQL4 (Figure 1A). From the String database, RECQs’ interacting proteins were identified. Next, the interaction network was built according to the diagram presented in Figure 1B. As a result, we discovered notable protein interactions involving BRCA1, EXO1, FANCM, FBN, MUS81, RAD51, TP53, and TOP3A. Subsequently, GO and KEGG enrichment analyses were performed on all the molecules that interacted. The analysis of Gene Ontology revealed that RECQs played a role in DNA replication, DNA repair, and the meiotic cell cycle. Furthermore, the KEGG pathway analysis indicated that RECQs were involved in pathways such as “Homologous recombination”, “Platinum drug resistance”, “Mismatch repair”, and “Cell cycle” (Figure 1C).

### 3.2. RECQs mRNA and Protein Expression in Various Cancer Types

To investigate the expression of RECQ family genes at the mRNA level, a merger of TCGA samples and GTEx samples was performed. Overall, RECQs exhibited significant upregulation in the majority of tumor tissues when compared to normal tissues, particularly in CHOL, DLBC, ESCA, GBM, HNSC, LGG, LIHC, PAAD, PRAD, SKCM, STAD, TGCT, THCA, and THYM (Figure 2A). Following the removal of normal GTEx samples, RECQs continued to be upregulated in CHOL, ESCA, HNSC, LIHC, and STAD (Appendix A). Subsequently, by utilizing matched samples of tumor tissues and adjacent normal tissues from the TCGA dataset, it was observed that the majority of RECQ family genes exhibited increased expression levels in BRCA, COAD, LIHC, LUSC, and STAD (Figure 2B). Furthermore, the TCGA dataset was applied to assess RECQ mRNA expression levels in pan-cancers. In the figure, RECQL4 exhibited the highest level of expression, while BLM showed a comparatively lower overall expression (Figure 2C). In addition, the HPA database showed that BLM and RECQL4 had moderate to strong staining in BLCA, CESC, COAD, GBM, HNSC, OV, PAAD, SKCM, and THCA, while RECQs (excluding RECQL5) protein levels were expressed. In the meantime, the expression of RECQL was relatively high in KIRC, SKCM, TGCT, and THCA (Figure 2D).

### 3.3. Variations in Genetic Alterations and Methylation of RECQs in Different Cancers

Genomic and epigenetic alterations impact both tumor formation and immune tolerance. The mutation status of RECQs in various tumors was examined using cBioPortal. Based on the results, it has been shown that RECQL, BLM, RECQL4, and RECQL5 have relatively high variation frequencies. However, the presence of RECQs was infrequent in essential thrombocythemia, mature B-cell neoplasms, myelodysplastic neoplasms, and acute myeloid leukemia. Significantly, every RECQ exhibited substantial modification, particularly in BLCA and BRCA (Figure 3A). In every cancer type, the percentage of CNV and its impact on RECQ expression were investigated. A greater proportion of CNV was detected in TGCT, SKCM, LUCS, ACC, ESCA, STAD, HNSC, UCS, OV, BLCA, LIHC, and COAD. The alteration trend of RECQL, RECQL4, and RECQL5’s heterozygous/homozygous CNV (deletion/amplification) status exhibited a resemblance to that of BLM and WRN (Figure 3B). As illustrated in the accompanying Appendix A, there were observed positive correlations between the expression of RECQs and CNV in most types of cancer. Afterward, we conducted a correlation analysis to examine the effect of DNA methylation in the promoter region on the expression of RECQs (Figure 3C). Methylation (−1 < cor < −0.6) significantly suppressed the expression of mRNA level of BLM in TGCT and LIHC, WRN in TGCT, RECQL4 in TGCT, UVM, and OV, and RECQL5 in BLCA, UVM, THYM, and OV. Appendix A displayed the most important tumor data and methylation correlation for each member of the RECQ family, comprising five data points. The variation in methylation between tumor and normal samples was presented in Figure 3D. In BRCA, WRN exhibited elevated methylation levels, while in COAD and HNSC, RECQL5 showed increased methylation. Furthermore, RECQs had lower methylation levels in normal tissues in most cancers. The results indicate that genetic and DNA methylation alterations may be responsible for the abnormal expression of RECQs in certain types of cancer.

### 3.4. The Correlation between RECQs and Immune Infiltration in Pan-Cancer

The precise mechanism by which RECQs control the tumor microenvironment (TME) remains unidentified. Considering that stromal and immune cells are the main effectors of TME, the relationship between RECQs and stromal and immune cell scores was evaluated in pan-cancer (Figure 4A,B). BLM, WRN, RECQL4, and RECQL5 were repressed by stromal and/or immune elements in most types of cancer, while RECQL exhibited reverse patterns. Moreover, the stromal and immune scores exhibited a comparable pattern for a single gene across the majority of cancers. In THYM, the stromal score of RECQs showed an inverse relationship with the immune score (Figure 4A,B). RECQs, with the exception of RECQL, exhibited a negative correlation with non-cancer components in most cancer types, as inferred from the comprehensive ESTIMATEscore (Figure 4C). Hence, the TIMER algorithm was employed to thoroughly assess RECQs and the infiltration of key immune cells (such as T cells, neutrophils, myeloid dendritic cells, and macrophages), aligning with the ESTIMATEscore pattern. RECQs were expressed at high levels in immune subtypes C1 and C2, with RECQL exhibiting the highest expression level in subtype C6. On the other hand, BLM and RECQL4 demonstrated the lowest expression levels in isoform C3 (Figure 5A). Moreover, the expression of RECQs in each cancer subtype yielded the top five noteworthy variations in data (Appendix A). According to the RNA expression data, RECQL and WRN demonstrated a negative correlation with stem cell scores across the majority of cancers. Conversely, BLM, RECQL4, and RECQL5 displayed diverse correlations with stem cell scores across different types of cancers. Figure 5B showed that as the tumor stemness index score increases, the activity of tumor stem cells also increases, while the degree of differentiation decreases. In most cancers, the DNA methylation data indicated a negative correlation between RECQL and tumor stemness, while different family members showed contrasting results in various cancers (Figure 5C). The therapeutic outcomes and prognosis of cancer immunotherapy are linked to TMB and MSI [35,36,37]. The correlation between RECQs mRNA expression and TMB/MSI was assessed in Figure 5D. As a result, RECQL4 exhibited a significant correlation with TMB/MSI in the majority of cancers. RECQs were associated with TMB/MSI specifically in the cases of COAD, STAD, and UCEC. Collectively, there is potential for RECQs to serve as molecules with immunotherapy functions.

### 3.5. The Relationship between mRNA Expression and the Predictive Significance of RECQs

Through analysis of TCGA data, we have identified a correlation between the expression of RECQs messenger RNA and the prognosis of patient survival, including OS, DSS, DFI, and PFI. In the majority of cancer cases, a negative prognosis was indicated by elevated levels of RECQs expression. However, several notable exceptions were identified, namely RECQL and WRN in KIRC, and BLM in STAD and OV, all of which exhibited a discernible protective function (Figure 6A). Additionally, we presented a comprehensive forest map of OS using Cox regression in TCGA, revealing that RECQs have unfavorable prognoses in LGG and LIHC (Appendix A).

### 3.6. The Relationship between the Expression of RECQs and the Clinical Characteristics in LIHC

The results indicated that RECQ helicases, particularly RECQL4, had a significant impact on LIHC. In order to assess the diagnostic effectiveness of RECQs in LIHC tissues and normal samples, the ROC curve was employed. WRN exhibited poor accuracy in predicting both normal and tumor outcomes (AUC = 0.575, CI = 0.526–0.625). On the other hand, RECQL4 demonstrated the highest accuracy (AUC = 0.911, CI = 0.884–0.937) (Figure 6B). The prognosis value of RECQs was predicted using Cox regression univariate analysis, in conjunction with clinical characteristics. Therefore, the prognostic significance of LIHC was significantly influenced by factors such as pathological stage, tumor status, and the expression levels of RECQL, WRN, RECQL4, and RECQL5 (Table 1). The variables that showed significant statistical significance in the univariate analysis (*p* < 0.05) were subsequently incorporated into the multivariate analysis. Table 1 also showed that LIHC was influenced by the pathological stage (HR = 2.075, *p* < 0.001), tumor status (HR = 1.794, *p* = 0.004), and RECQL4 (HR = 1.554, *p* = 0.031), which acted as separate predictors. A prognostic model was created for LIHC using Cox regression analysis, incorporating pathological stage, tumor status, and RECQL4 expression. Following that, we showcased the forecast of 1-year, 3-year, and 5-year OS in the form of a nomogram (Figure 6C). The model effectively forecasted the real-life 1-, 3- and 5-year survival rates by visually examining the alignment between the observed probability and the model-predicted probability under varying conditions (Figure 6D). Furthermore, we examined the predictive significance of RECQL4 in a subgroup of LIHC individuals (Figure 6E). As a result, RECQL4 was found to be associated with a negative prognosis for OS in patients with pathologic stage (stage I and II) (HR = 1.63, *p* = 0.044) and (stage III and IV) (H = 1.85, *p* = 0.031), specifically in males (HR = 1.87, *p* = 0.005), individuals below the age of 60 (HR = 2.56, *p* = 0.002), those with a body mass index (BMI) (HR = 2.13, *p* = 0.003), R0 resection (HR = 1.74, *p* = 0.004), and those with tumor status (HR = 2.22, *p* = 0.001).

### 3.7. Preliminary Verification of Characteristics of RECQL4 in LIHC

We targeted the expression of RECQL4 in LIHC, and its clinical significance as well as prognostic value in the GEO database to confirm the above findings. Using the CCLE database, we first looked at RECQL4 expression at the transcriptome level in LIHC cell lines (Figure 7A). The HPA database was used to determine the subcellular localization of RECQL4 protein in A431 and U251MG cells using ICC, and the results indicated that RECQL4 was primarily expressed in the nucleoplasm (Figure 7B). To confirm the dysregulation of RECQL4, a histochemical analysis was performed in LIHC and peritumoral tissues in the HPA database (Figure 7C). Further, with a testing dataset in GEO, RECQL4 expression was determined in nine pairs of LIHC and peritumoral tissues. The findings showed higher RECQL4 expression in most tumor tissues (Figure 7D). By using the Kaplan–Meier plotter database, patients with high RECQL4 expression had shorter OS (HR = 1.9, *p* = 0.00021) (Figure 7E).

## 4. Discussion

The aim of this research was to investigate the possible involvement of RECQ family members in various types of cancer. Significant variations in mRNA and protein levels were observed among different types of cancer in RECQs. Earlier findings suggested that the RECQs expression showed notable differences between tumors and non-neoplastic conditions [38]. In tumors, somatic chromosomal deletions and replications are characterized by CNVs, indicating that it may serve as a fertile environment for acquired alterations in cancer cells [39]. Furthermore, the expression of RECQs is controlled by epigenetic mechanisms, and gene expression can be modified by DNA methylation without affecting the gene sequence [40]. The investigation of RECQs changes at the genomic and epigenetic levels revealed that CNV and promoter DNA methylation play a role in modulating RECQs expressions in different types of cancers. Our hypothesis suggests that the expression of RECQs may be influenced to a certain degree by the evolution of tumor cells, and these genetic alterations have previously been associated with malignant tumor behavior and the development of cancer.

The RECQ family is associated with the functions of recombination, replication, and DNA repair, as indicated by GO and KEGG enrichment analyses. According to prior research, RECQL facilitates the progression of non-small cell lung carcinoma through the regulation of epithelial–mesenchymal transition [41]. BLM facilitates the growth of invasive breast cancer by enhancing the duplication of DNA and the multiplication of cells [42]. The stability of the cancerous cell genome is preserved by the elevated expression of WRN, while the suppression of WRN leads to the demise of the cell [43]. Tumor growth is diminished when RECQL4 expression is disrupted in osteosarcoma and prostate cancer cells, where RECQL4 is abundantly present [44]. RECQL5 has the ability to act as a suppressor of osteosarcoma tumors and could be a promising target for osteosarcoma treatment [45]. Nevertheless, the excessive expression of RECQL5 stimulates cellular growth and contributes to the progression of bladder carcinoma [46]. The role of RECQs in different cancers, whether as an oncogene or a tumor suppressor, is still uncertain, and the precise underlying mechanisms are not yet known.

Furthermore, we investigated the correlation between RECQs and immune modulation. The intricate tumor microenvironment consists of cancer cells, immune cells, stromal cells, and extracellular components, each playing diverse functions. Certain positions play a part in creating a suppressive atmosphere that impacts different tumor treatment outcomes [47]. According to our study, stromal and immune scores, as well as an ESTIMATE score, were computed for RECQs in various types of cancer. Furthermore, it was found that RECQs are associated with inhibitory immune factors in different types of tumors, elucidating the reason behind the unfavorable prognosis observed in cancer patients expressing RECQs. Moreover, the varying associations of RECQs in immune subcategories may impact the variation in cancer prognosis. Tumor-propagating cells, alternatively referred to as cancer stem cells, possess the ability to renew themselves and maintain an undifferentiated state, making them impervious to chemoradiotherapy [48]. Cancer stemness was determined by utilizing RNAss, which is based on mRNA expression, and DNAss, which is based on DNA methylation [49]. In 33 cancers, we discovered a noteworthy association between RECQs and stem cell scores. The expression of RECQs in THCA is inversely correlated with the stem cell score, suggesting that higher RECQ expression levels are linked to a reduction in the number of tumor stem cells and an increase in tumor differentiation. Furthermore, according to DNAss, the RECQ expression in DLBC, LUAD, and TGCT showed a positive correlation with the stem cell score, suggesting that a higher level of RECQ expression is associated with increased tumor stem cell characteristics and decreased differentiation. The properties of stem cells in gastric cancer are regulated by WRN [50]. We observed RECQ expression was linked with TMB and MSI in some cancer types. Cancers exhibiting elevated TMB levels have a greater likelihood of producing immunogenic peptides, consequently impacting the effectiveness of immunotherapy. Genomic instability, believed to be caused by high frequencies of MSI, was considered a catalyst for the onset of cancers [51]. A new investigation has discovered a hopeful synthetic lethal connection between disabling/restraining the WRN DNA helicase and colorectal cancer with MSI, a characteristic that emerges due to a lack of DNA mismatch repair [52]. In the meantime, the loss of BLM function due to MSI can enhance the genetic instability of an already unstable genotype in gastric tumors [53]. In our research, we also discovered consistent findings, and RECQL4 exhibited a robust correlation with TMB and MSI in numerous types of cancer. Consequently, RECQs were potential targets for immunotherapeutic strategy.

Newly available data indicated that TMB had the potential to serve as an indicator for immune checkpoint inhibitors in different types of cancer. Tumors with a significant number of mutations encourage the development of new antigens, enhancing their immunogenicity and boosting the effectiveness of immunotherapy [54,55]. The MSI can occur due to the removal or addition of base pairs in the microsatellite region, making it a possible indicator for predicting the effectiveness of tumor immunotherapy. Immunotherapy yields superior outcomes for tumors exhibiting mismatch repair deficiency (dMMR) and MSI-H [56]. In this study, the expression of BLM and WRN in COAD showed a positive correlation with both TMB and MSI, while RECQL5 exhibited a negative correlation with both. Moreover, nearly all RECQs exhibited a positive correlation with the two targets in UCEC. In the selection of patients eligible for immunotherapy in various cancers, the utilization of TMB and MSI analyses is crucial [57].

Moreover, it was found that RECQs exhibited a notable variation in prognosis across different types of cancers, with RECQL, BLM, and RECQL4 showing promising diagnostic potential in LIHC. RECQs have a detrimental prognostic impact on the majority of cancers according to the sequencing data, while indicating a more favorable prognosis in a few specific tumor categories. Tumor treatment faces a significant challenge due to the diversity of cancers, which can explain the varying importance of a specific factor in predicting outcomes [58,59]. Earlier investigations indicated that increased RECQL1 expression was associated with poorer OS and PFI in patients with LGG [14]. On the other hand, ER-positive breast cancer patients who had a RECQL protein level above the 75th percentile experienced improved DSS over a 15-year period, particularly among women [60]. Increased expression of BLM is associated with decreased OS in bladder cancer and reduced DFI in cholangiocarcinoma [13,61]. Breast cancer patients with high RECQL5 expression have a negative prognosis [62]. Lower OS is linked to increased expression of RECQL, BLM, and WRN in pancreatic cancer [63,64,65]. However, there has been limited research on the clinical significance of RECQL4 in LIHC. The connection between RECQL4 levels in the TCGA databases and a decreased OS of LIHC patients was established in this study. Analysis of the clinical subtype also indicated that RECQL4 was a separate adverse prognostic determinant for LIHC linked to the pathological stage. This work focuses on the central role of RECQL4 in promoting and suppressing tumors in different types of cancer. Hence, additional empirical investigations are required to uncover the mechanism through which RECQL4 facilitates the malignant activity of LIHC cells, leading to an unfavorable prognosis for patients.

There are significant constraints in this research. Initially, we investigated the involvement of RECQ family members in pan-cancer from a comprehensive standpoint; nevertheless, further experimental and clinical verification is imperative for molecules holding promising research significance. We will collect tumor tissue and survival data from LIHC patients and correlate them with RECQL4. In addition, a complete description of the corresponding mechanisms is lacking. We will explore the specific mechanisms of RECQL4 pathogenesis in LIHC cell lines. Thirdly, the unavailability of expression data for immunotherapy patients makes it challenging to anticipate and assess the effectiveness of immunotherapy. We will address these concerns in more detail in our future investigations.

To summarize, we employed a multi-omics strategy to investigate the mRNA and protein levels, possible roles, immune cell infiltration, clinical characteristics, and predictive significance of the RECQ gene group. Additionally, we have verified that the expression of RECQL4 is linked to an unfavorable prognosis in LIHC. The results validate the importance of RECQ expression in forecasting tumor prognosis within the immune microenvironment and offer crucial points of reference for future investigations.

## Figures and Tables

**Figure 1 biomedicines-11-02318-f001:**
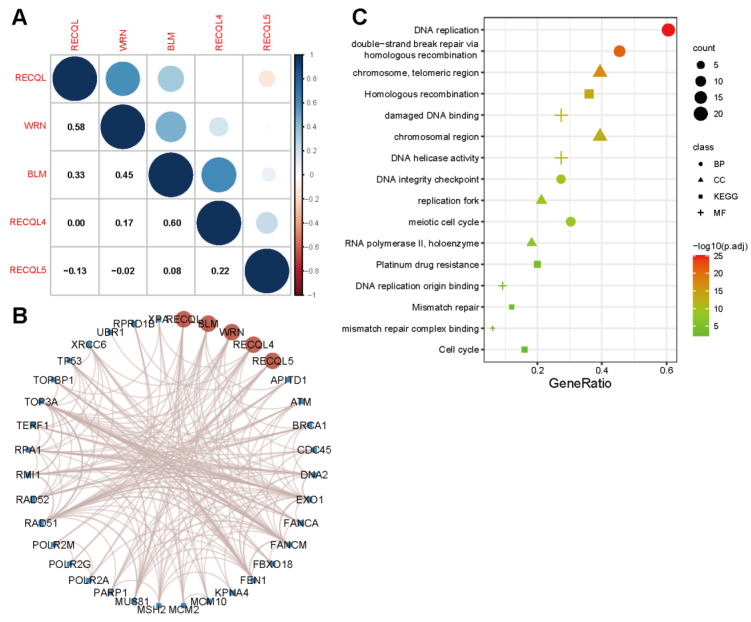
RECQ family genes and function prediction. (**A**) A correlation between mRNA expression in RECQs. (**B**) The PPI network analysis of RECQs was exhibited from String. (**C**) Analysis of GO and KEGG enrichments by interacting proteins.

**Figure 2 biomedicines-11-02318-f002:**
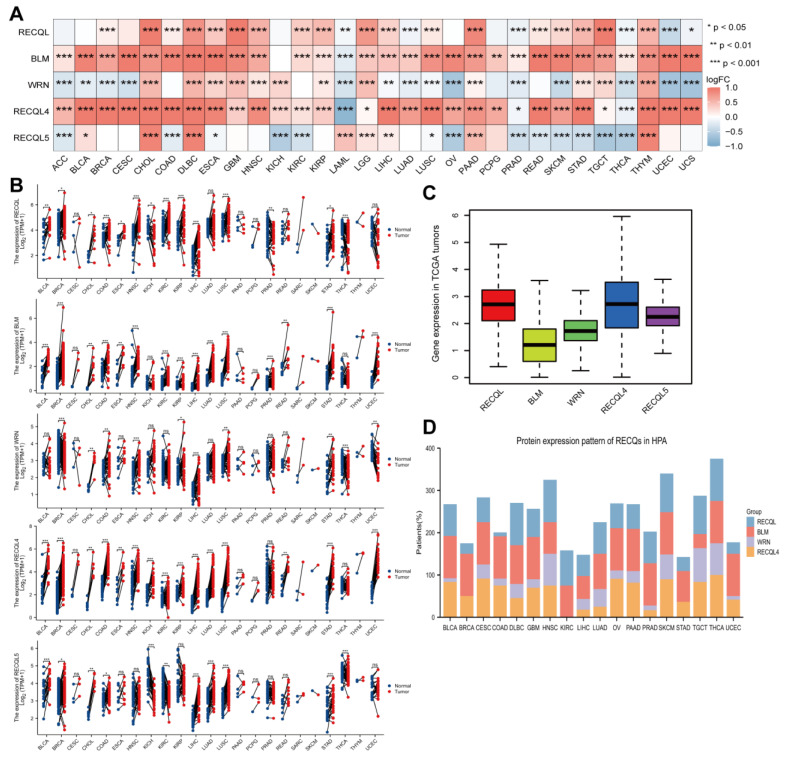
Expression level of RECQ family in pan-cancer. (**A**) GTEx and TCGA were used to assess the differential mRNA expression of RECQs in tumor and normal tissues. (**B**) TCGA paired cancer and para-cancer mRNA expression of RECQs. (**C**) Expression of RECQs at the total mRNA level in pan-cancer. (**D**) Variations in the expression of RECQ proteins among different HPA tumors. * *p* <  0.05, ** *p* < 0.01, *** *p* < 0.001. ACC, adrenocortical carcinoma; BLCA, bladder urothelial carcinoma; BRCA, breast invasive carcinoma; CESC, cervical squamous cell carcinoma and endocervical adenocarcinoma; CHOL, cholangiocarcinoma; COAD, colon adenocarcinoma; DLBC, lymphoid neoplasm diffuse large B-cell lymphoma; ESCA, esophageal carcinoma; GBM, glioblastoma multiforme; HNSC, head and neck squamous cell carcinoma; KICH, kidney chromophobe; KIRC, kidney renal clear cell carcinoma; KIRP, kidney renal papillary cell carcinoma; LAML, acute myeloid leukemia; LGG, brain lower-grade glioma; LIHC, liver hepatocellular carcinoma; LUAD, lung adenocarcinoma; LUSC, lung squamous cell carcinoma; MESO, mesothelioma; OV, ovarian serous cystadenocarcinoma; PAAD, pancreatic adenocarcinoma; PCPG, pheochromocytoma and paraganglioma; PRAD, prostate adenocarcinoma; READ, rectum adenocarcinoma; SARC, sarcoma; SKCM, skin cutaneous melanoma; STAD, stomach adenocarcinoma; TGCT, testicular germ cell tumors; THCA, thyroid carcinoma; THYM, thymoma; UCEC, uterine corpus endometrial carcinoma; UCS, uterine carcinosarcoma; UVM, uveal melanoma.

**Figure 3 biomedicines-11-02318-f003:**
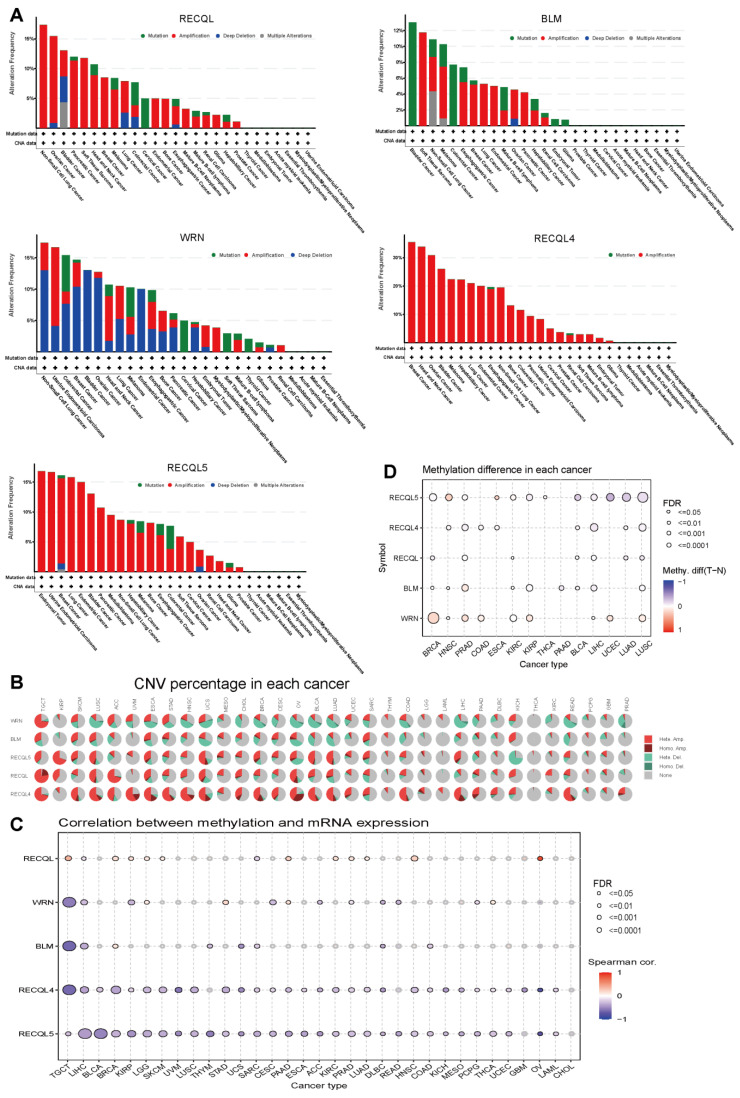
RECQ family variations and DNA methylation in pan-cancer. (**A**) cBioPortal analyzed the alteration frequency of RECQs in different cancers. (**B**) GSCA evaluated the CNV alterations of RECQs in cancer. (**C**) TCGA investigated the correlation between DNA methylation and RECQ mRNA expression. (**D**) Samples paired with tumor and para-cancer have different methylation patterns.

**Figure 4 biomedicines-11-02318-f004:**
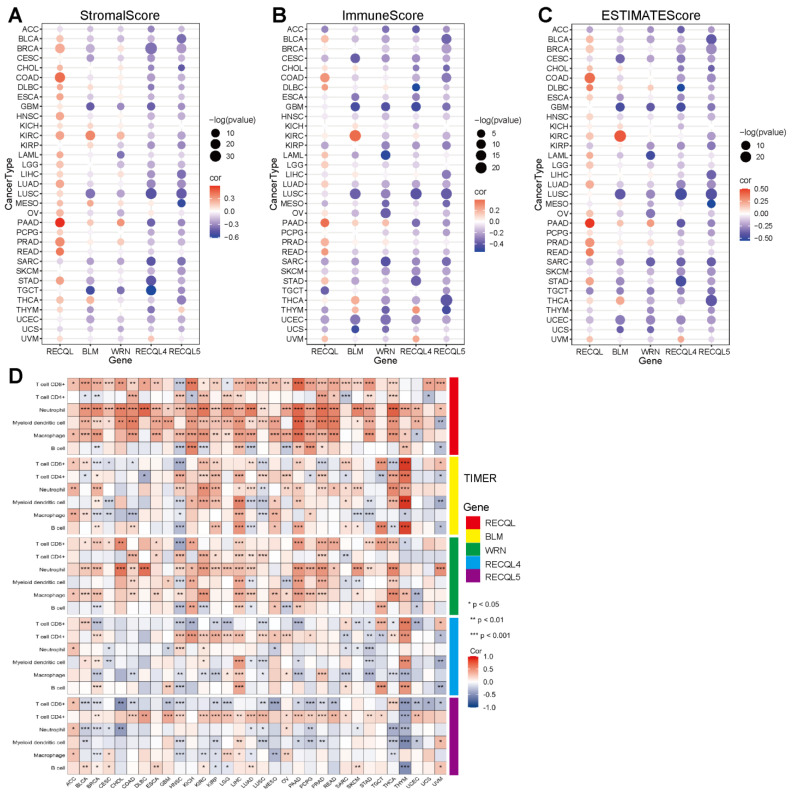
A correlation analysis of RECQ family expression and pan-cancer microenvironment. (**A**–**C**) To analyze the correlation between RECQs and StromalScores, ImmuneScores, and ESTIMATEScores. (**D**) An analysis of the correlation between RECQs mRNA expression and immune cell types at TCGA by TIMER. * *p* < 0.05, ** *p* < 0.01, *** *p* < 0.001.

**Figure 5 biomedicines-11-02318-f005:**
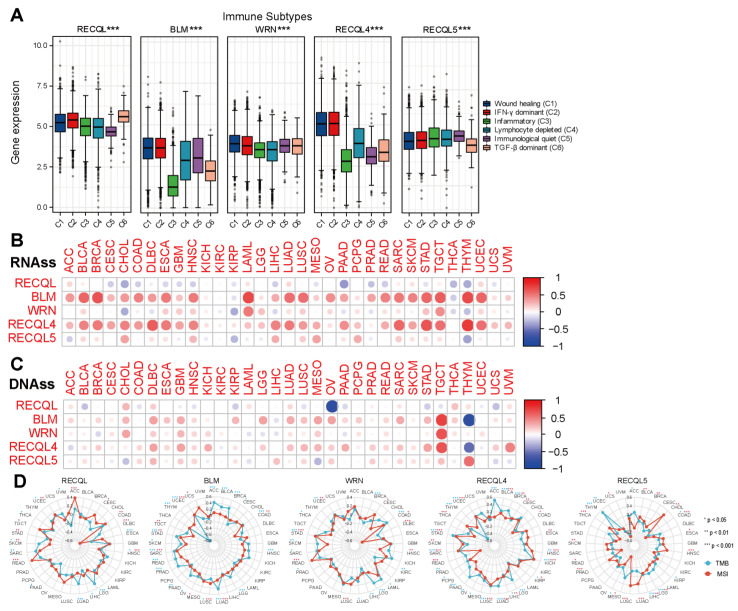
RECQ family expression correlated with immune subtypes, RNAss/DNAss, and TMB/MSI. (**A**) Expression of RECQ mRNA among pan-cancer immune subtypes. (**B**,**C**) To analyze the correlation between RECQs and tumor stemness by RNAss/DNAss. (**D**) To analyze the correlation between RECQs and TMB/MSI by Radar diagram. * *p* < 0.05, ** *p* < 0.01, *** *p* < 0.001.

**Figure 6 biomedicines-11-02318-f006:**
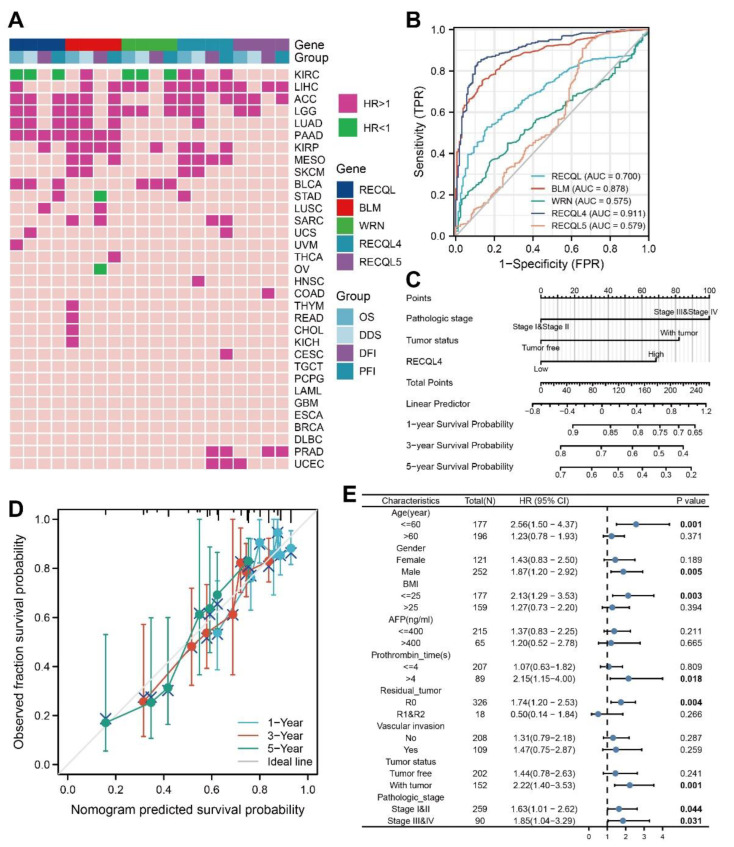
The clinical characteristics of RECQs in pan-cancers. (**A**) Correlation between RECQs mRNA expression and OS, DSS, DFI, and PFI from TCGA. (**B**) ROC curves for RECQs in LIHC samples and normal samples by TCGA and GTEx. (**C**) A nomogram to predict one-year, three-year, and five-year OS for patients with LIHC. (**D**) Nomogram calibration for Cox regression models and fitting analysis of actual situations. (**E**) RECQL4 expression in different subgroups assessed by univariate survival analysis.

**Figure 7 biomedicines-11-02318-f007:**
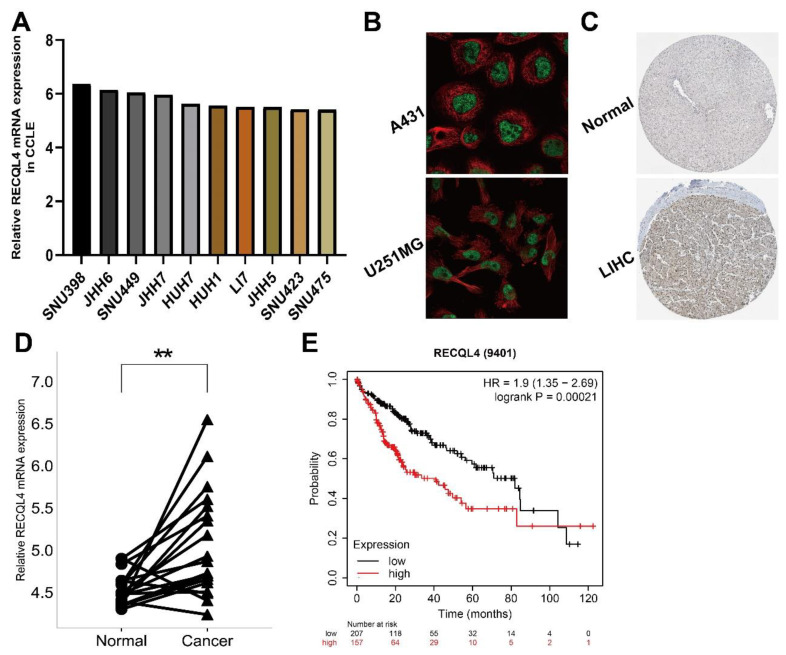
Preliminary verification of characteristics of RECQL4. (**A**) RECQL4 mRNA expression level in LIHC cell lines from CCLE. (**B**) In A431 and U251MG cell lines, HPA immunocytochemistry was used to determine the subcellular location of RECQL4, which localized to the nucleoplasm (green). (**C**) The IHC pictures of RECQL4 in normal and tumor tissues analyzed by the HPA. (**D**) The RECQL4 mRNA expression on 19 LIHC and paraneoplastic paired tissues was verified by GSE14520 dataset. (**E**) Overall survival analysis of 364 LIHC patients according to RECQL4 expression was performed by Kaplan–Meier plotter database. ** *p* < 0.01.

**Table 1 biomedicines-11-02318-t001:** Analysis of overall survival in TCGA patients with LIHC by univariate and multivariate methods.

Characteristics	Total (N)	Univariate Analysis		Multivariate Analysis	
		Hazard Ratio (95% CI)	*p* Value	Hazard Ratio (95% CI)	*p* Value
Age	373				
≤60	177	Reference			
>60	196	1.205 (0.850–1.708)	0.295		
Gender	373				
Female	121	Reference			
Male	252	0.793 (0.557–1.130)	0.2		
Race	361				
White	185	Reference			
Asian and Black or African American	176	0.791 (0.551–1.135)	0.203		
BMI	336				
≤25	177	Reference			
>25	159	0.798 (0.550–1.158)	0.235		
Tumor status	354				
Tumor free	202	Reference			
With tumor	152	2.317 (1.590–3.376)	**<0.001**	1.794 (1.200–2.684)	**0.004**
Residual tumor	344				
R0	326	Reference			
R1 and R2	18	1.604 (0.812–3.169)	0.174		
Pathologic stage	349				
Stage I and Stage II	259	Reference			
Stage III and Stage IV	90	2.504 (1.727–3.631)	**<0.001**	2.075 (1.393–3.091)	**<0.001**
Adjacent hepatic tissue inflammation	236				
None	118	Reference			
Mild and Severe	118	1.194 (0.734–1.942)	0.475		
AFP (ng/mL)	279				
≤400	215	Reference			
>400	64	1.075 (0.658–1.759)	0.772		
Albumin (g/dL)	299				
<3.5	69	Reference			
≥3.5	230	0.897 (0.549–1.464)	0.662		
Prothrombin time	296				
≤4	207	Reference			
>4	89	1.335 (0.881–2.023)	0.174		
Vascular invasion	317				
No	208	Reference			
Yes	109	1.344 (0.887–2.035)	0.163		
RECQL (high vs. low)	373	1.454 (1.026–2.060)	**0.035**	1.047 (0.637–1.719)	0.858
BLM (high vs. low)	373	1.270 (0.900–1.793)	0.174		
WRN (high vs. low)	373	1.473 (1.041–2.086)	**0.029**	1.235 (0.752–2.027)	0.405
RECQL4 (high vs. low)	373	1.672 (1.180–2.371)	**0.004**	1.554 (1.042–2.318)	**0.031**
RECQL5 (high vs. low)	373	1.554 (1.098–2.199)	**0.013**	1.221 (0.825–1.809)	0.318

## Data Availability

The datasets supporting the conclusions of this article are available in the GTEx and TCGA database (https://ngdc.cncb.ac.cn/databasecommons/database (accessed on 1 March 2023)), (https://www.cancer.gov/ccg/research/genome-sequencing/tcga (accessed on 1 March 2023)).

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
