# Peer review of "The Expression Characteristics and Function of the RECQ Family in Pan-Cancer"

_biomedicines, 2023, doi:10.3390/biomedicines11082318_

Round 1
Reviewer 1 Report
This article reports the expression characteristics and function of the RECQ family in pan-cancer. This research is aimed to understand the role of RECQ in progression of different cancers. This article is well designed to discover the significance of RECQ in tumor formation and supported with various analytical approaches. This article should be considered for acceptance after the minor correction.
1) There is a necessity to improve the label of x and y axis in Figure 2(B), Figure 3(A).
2) Authors should briefly suggest the strategies to overcome the limitations of this study in the discussion section.
Need minor checking.
Reviewer 2 Report
The authors of the original article performed a multi-omics analysis of RecQ helicase family members in 33 cancer types. Their aim was to better understand the expression of RecQ family members, their role in tumorigenesis, stem cell phenotype maintenance, and TME immune regulation. They also analyzed the prognostic role of the RecQ family.
Among other things, they found that RecQ family elements show mutational, gene expression, and epigenetic differences in all tumors studied and are also involved in TME immune regulation, although of course differently from tumor type to tumor type. They also highlight an independent prognostic role for RecQL4 in liver cancer.
The study is a very well-thought-out and well-constructed in-silico bioinformatic analysis. It used the TCGA, GTEx, cBioPortal, GSCA, GO, KEGG, and TISIDB databases and complex bioinformatics and biostatistical methods to derive the results.
In themselves, the methods used and the results obtained are understandable, and there is nothing to complain about.
However, from a clinical or biological point of view, the study is incomplete.
Some of the data obtained by in silico bioinformatics analysis should be validated, at least in cell culture experiments or by including a small number of independent human samples.
The data obtained from statistical tests are very promising, and since they are derived from the analysis of data from human samples, we tend to believe that the resulting correlations would work in the same way in an independent model or on a human sample set. This is conceivable in the majority of cases, but to prove this and thus validate the results obtained, I would consider it justified to at least investigate the association of RecQL4 with liver cancer (e.g., by analyzing survival data from 8–10 patients with liver cancer and by examining RecQL4 expression in histopathological samples; or by examining in cell culture models the influence of the RecQ family members on cancer stem cell phenotypes).
A major revision is required before accepting the manuscript for publication.
Round 2
Reviewer 2 Report
On the basis of the amendments made by the authors and their detailed replies, I find the revised manuscript acceptable for publication.